# Species-wide gene editing of a flowering regulator reveals hidden phenotypic variation

**Ulrich Lutz**[1¤a]*, **Ilja Bezrukov**[1], **Rebecca Schwab**[1], **Wei Yuan**[1], **Marius Kollmar**[1¤b], **Detlef Weigel**[1,2]*

**1** Department of Molecular Biology, Max Planck Institute for Biology Tübingen, Tübingen, Germany,
**2** Institute for Bioinformatics and Medical Informatics, University of Tübingen, Tübingen, Germany

¤a Current address: Biogenda e.U., Tübingen, Germany
¤b Current address: Department of Physiology of Yield Stability, Faculty of Agriculture, University of Hohenheim, Stuttgart, Germany
* ulrich.lutz@tuebingen.mpg.de (UL); weigel@tue.mpg.de (DW)

## Abstract

Genes do not act in isolation, and the effects of a specific variant at one locus can often be greatly modified by polymorphic variants at other loci. A good example is *FLOWERING LOCUS C* (*FLC*), which has been inferred to explain much of the flowering time variation in *Arabidopsis thaliana*. We use a set of 62 *flc* species-wide mutants to document pleiotropic, genotype-dependent effects for *FLC* on flowering as well as several other traits. Time to flowering was greatly reduced in all mutants, with the remaining variation explained mainly by allelic variation at the *FLC* target *FT*. Analysis of *FT* sequence variation suggested that extremely early combinations of *FLC* and *FT* alleles should exist in the wild, which we confirmed by targeted collections. Our study provides a proof of concept on how pan-genetic analysis of hub genes can reveal the true extent of genetic networks in a species.

## Introduction

The appropriate onset of flowering is highly adaptive and greatly affects fitness. Flowering is orchestrated by a large network of genes integrating a multitude of endogenous and environmental signals, with *FLOWERING LOCUS C* (*FLC*) and its upstream regulator *FRIGIDA* (*FRI*) as critical components of this network in *Arabidopsis thaliana*. Together, *FLC* and *FRI* prevent flowering before the arrival of favorable spring conditions. During extended periods of sustained cold, *FLC* is gradually downregulated by a process called vernalization, allowing for the upregulation of flowering promoters such as *FLOWERING LOCUS T* (*FT*) once winter is over [1–5]. Given their central roles, it is not surprising that there is tremendous functional variation at *FRI* and *FLC* in *A. thaliana* populations. In contrast to *FRI*, complete inactivation of *FLC* in natural populations is, however, rare. Instead, *FLC* activity is typically reduced to different extents by intronic transposon insertions, with point mutations in

**Data availability statement:** RNA-seq and WGS reads were deposited at ENA with the accession numbers PRJEB89960 and PRJEB8996. The data underlying the figures can be found in S1 Data and https://doi.org/10.5281/zenodo.15403194. The TIGER snakemake pipeline is available at https://github.com/ibebio/tiger-pipeline and https://doi.org/10.5281/zenodo.15535778. All tools used for the bioinformatic analyses are publicly available. Unless specified otherwise, default parameters were used.

**Funding:** This work was supported by the Novo Nordisk Foundation (Novonesis Prize to DW) and the Max Planck Society (to DW). The Max Planck Society paid the salaries of all authors. The funders played no role in the study design, data collection and analysis, decision to publish, or preparation of the manuscript.

**Competing interests:** We have read the journal's policy and the authors of this manuscript have the following competing interests: UL is the founder of Biogenda, a company specializing in customized genetic diagnostics for the agricultural sector. DW holds equity in Computomics, which advises plant breeders. DW also consults for KWS SE, a globally active plant breeder and seed producer. All other authors declare no competing interests.

**Abbreviations:** CLN, cauline leaf number; DEGs, differentially expressed genes; DTF, distribution of flowering time; *FLC, FLOWERING LOCUS C*; *FRI, FRIGIDA*; *FT, FLOWERING LOCUS T*; LES, leaf economics spectrum; PRA, projected rosette area; QTL, quantitative trait loci; RLN, rosette leaf number; TIGER, trained individual genome reconstruction; TLN, total leaf number; WGS, whole-genome sequencing.

non-coding sequences additionally affecting the time and temperature required for full vernalization [5]. While other loci affect flowering time variation as well, their contributions are often masked due to the large effects of *FLC* and *FRI* alleles, especially for those that map close to *FLC* [6].

*FLC* is best known for its function in flowering, but it has also multiple pleiotropic activities in germination, vegetative development, circadian rhythmicity, and drought tolerance and even viral tolerance [7,8], consistent with binding of the FLC protein to promoters of hundreds of genes [9–11]. In light of so many traits being controlled by *FLC* it has become less clear whether *FLC*'s adaptive role should be primarily seen through the lens of flowering time. Instead, it has been proposed that *FLC* must be appreciated as a central nexus with broad contributions to the execution of diverse ecological strategies [12]. Because the relationship between gene activity and specific phenotypes is often not linear, specific alleles at pleiotropic loci may disproportionately affect some phenotypes compared to others, which raises interesting questions as to the optimal *FLC* activity in specific environments.

To dissect the broader pleiotropic roles of *FLC* and the hidden spectrum of *FLC*-independent variation in flowering time on a species-wide level, we have combined quantitative genetics with phenotypic, physiological, and transcriptomic studies of a species-wide collection of *FLC* knockdown and knockout alleles. We focus on experiments in controlled conditions to showcase the enormous potential of species-wide genetic disruptions for obtaining a holistic view of gene function.

## Results

### Hidden variation in flowering time revealed in *flc* mutants

Using an innovative pan-genetic approach – the editing of the same gene in different genetic backgrounds [13] – we previously established a collection of species-wide *flc* knockout (KO) and knockdown (KD) alleles [14] across a diverse set of 62 natural accessions. Overall, 84% of the *flc* mutant lines had reduced *FLC* transcript levels, with 39 likely having large deletions (KOs) and the rest having weaker mutations (KDs) [14].

This new set of lines allowed us to investigate how the effects of *FLC* mutations on flowering traits vary between many genetic contexts. To investigate the background-dependent effects of the mutations on flowering time we measured days to flowering (DTF), rosette leaf number (RLN) and cauline leaf number (CLN) under long days. In the mutants, broad-sense heritability ($H^2$) of the three traits was very similar (DTF: 0.95, RLN: 0.96, CLN: 0.95), while it was more varied and substantially lower in the wild types (DTF: 0.80, RLN: 0.67, CLN: 0.62).

All flowering time measurements were greatly reduced in the mutants compared to the corresponding wild types (Fig 1A). The correlation of *FLC* transcripts with RLN was also reduced in the mutants (Figs 1B, 1E and S2 and S1 Table). Notably, even complete *flc* KO lines varied considerably in flowering time (range DTF: 24.1, RLN: 26.7) (Fig 1B), with DTF < 16 and RLN < 8 in the earliest lines (Figs 1A, 1B and S1).

The later the wild types flowered, the more was flowering accelerated in the *flc* mutants. This was most strongly the case for DTF, but much less so for RLN and CLN. Some of this is likely due to non-genetic variance, but overall, it appears that

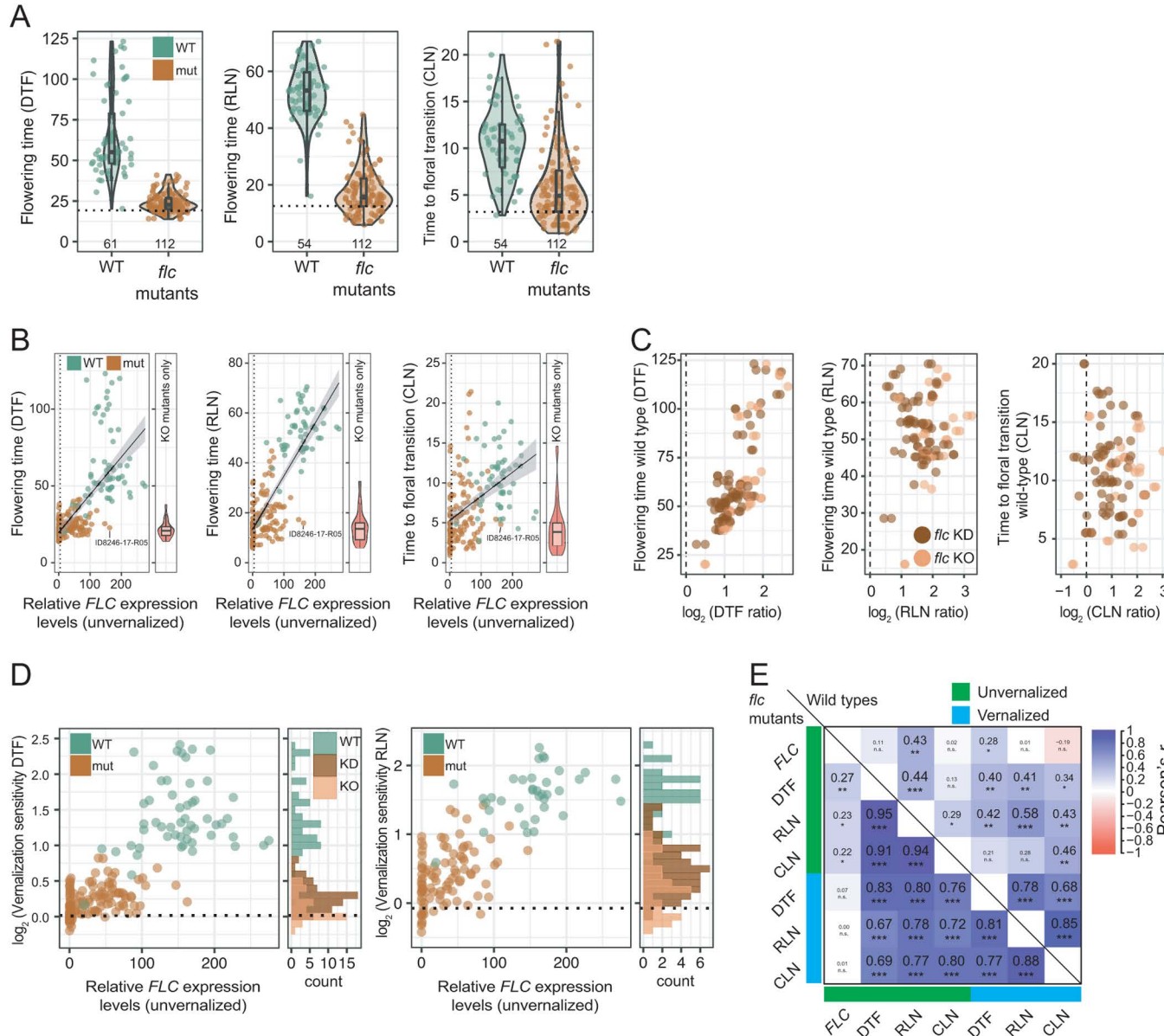

**Fig 1. Flowering time analysis of *flc* mutants. (A)** Flowering-related traits were measured at 22°C under LD, without vernalization. Means from 3 to 12 replicates per line are shown, with the number of measured lines indicated at the bottom of each graph. The black dashed horizontal line indicates Col-0. **(B)** Correlation between flowering times (S1 Table) and FLC expression levels (S2 Table) in unvernalized plants. Black dashed vertical line indicates an *FLC* relative expression level of 5 a.u., which divided the mutants into knockout (KO) and knockdown (KD) lines. To better illustrate the phenotypes of the KO mutants only, their values are shown as violin plots to the right in each subpanel. flc KO, DTF mean [±sd] 21.8 [±5.6], range 13.9 to 38.0, RLN 14.6 [±6.5], range 5.9 to 32.6. **(C)** Correlation of the mutant-versus-wild type phenotypic ratios (log$_2$[DTF$_{Wildtype}$/DTF$_{Mutant}$]) versus the wild-type phenotypic values. Left, DTF, Pearson's *r*, all lines *r* = 0.77, *p* < 2.2*e*−16, *df* = 108, KO only *r* = 0.81, *p* = 5.697*e*−10, *df* = 36. Middle, RLN, all lines *r* = 0.06, *p* = 0.58, *df* = 96; KO only *r* = 0.35, *p* = 0.037, *df* = 32. Right, CLN, all lines *r* = −0.017, *p* = 0.87, *df* = 96; KO only *r* = 0.28, *p* = 0.11, *df* = 32. Simple linear model values: multiple *r*$^2_{adj}$[*p*], log$_2$(DTF ratio KO) − log(*FLC*$_{Wildtype}$): 0.30 [0.0002], log$_2$(RLN ratio KO) − log(*FLC*$_{Wildtype}$): 0.18 [0.0068], log$_2$(CLN ratio KO) − log(*FLC*$_{Wildtype}$): 0.02 [>0.05]. The dashed line indicates 0. **(D)** Correlation between vernalization sensitivity (log$_2$) after eight weeks of vernalization and *FLC* expression levels before vernalization. Black dashed horizontal line indicates Col-0. Mutants: log$_2$(vernalization sensitivity DTF): mean [±sd] 0.23 [±0.21], range −0.20 to 0.82; log$_2$(vernalization sensitivity RLN): mean [±sd] 0.40 [±0.40], range −0.42 to 1.42; wild types: log$_2$(vernalization sensitivity DTF): mean [±sd] 1.41 [±0.49], range 0.17 to 2.41; log$_2$(vernalization sensitivity RLN): mean [±sd] 1.57 [±0.35], range 0.59 to 2.26; KO mutants only: log$_2$(vernalization sensitivity DTF); mean [±sd] = 0.12 [±0.19]), range: −0.20 to 0.5; log$_2$(vernalization sensitivity RLN): mean [±sd] = 0.20 [±0.38]), range: and −0.42 to 0.97. The distribution of vernalization sensitivities, as shown by the histograms, was analyzed separately for KO and KD populations. **(E)** Pearson's *r* of flowering traits before and after vernalization. *, *p* ≤ 0.05; **, *p* ≤ 0.01; ***, *p* ≤ 0.001; n.s., not significant. The data underlying this figure can be found in S1 Data and https://doi.org/10.5281/zenodo.15403194.

*FLC* affects not only time to bolting, the elongation of the main shoot, but also the leaf initiation rate (plastochron) (Figs 1C and S2). The considerably longer time to flowering in the wild types led to a larger spread in measurements (as stochastic environmental differences could act over a longer period), and also made precise measurements, especially of leaf number, more difficult, which together is reflected in the lower heritabilities for wild types compared to *flc* mutants. Taken together, we find that the effects of *flc* mutations on flowering vary strongly across genetic backgrounds.

While *FLC* is the major factor in vernalization of *A. thaliana*, vernalization has also an *FLC*-independent component [15,16]. We therefore measured vernalization sensitivity of DTF and of RLN. As expected, *flc* mutants were much less sensitive to vernalization than the wild types (Mann–Whitney *U*-test [MWU] for both DTF and RLN, $p < 0.0001$). Also as expected, residual vernalization sensitivities (expressed as $\log_2$ ratios of flowering time without and with vernalization) were higher for the KD lines than for the on average even earlier flowering complete KOs (Mann–Whitney *U* rank test, $p < 0.001$ for both DTF and RLN) (Fig 1D and S1 Table). Thus, modification of *FLC* activity might provide a simple means for fine-grained adjustment of vernalization sensitivity. The range of residual vernalization sensitivities in the knockout lines presents an opportunity to further dissect the basis of *FLC*-independent vernalization.

Across *A. thaliana* accessions, bolting tends to correlate with the time to initiation of the first flower. This translates into DTF/RLN and CLN usually changing in lock step, although this link can be genetically uncoupled [17]. We evaluated the contribution of *FLC* to linking DTF/RLN and CLN before and after vernalization, which greatly reduces *FLC* activity in wild types, by comparing the relationships between the flowering traits. In the unvernalized wild types, correlations between DTF, RLN, and CLN were low (range of Pearson's *r*: 0.13 to 0.44), but they increased with vernalization to a range (0.68 to 0.85) similar to the one in *flc* mutants regardless of vernalization treatment (0.67 to 0.95) (Figs 1E and S4). We conclude that *FLC* is a major component of trait de-canalization, or uncoupling, of flowering traits across natural genetic backgrounds and that vernalization efficiently canalizes flowering behavior through repression of *FLC*.

## The *FLC*-independent genetic architecture of extremely early flowering

In *A. thaliana*, the mapping of quantitative trait loci (QTL) for flowering time variation has predominantly involved crosses where at least one parent had strong *FLC* activity [18–29]. We used our collection of *flc* mutants – which could be considered artificial summer-annual lines – to ask whether complete removal of *FLC* can uncover new aspects of the genetic architecture of flowering time variation in *A. thaliana*. We generated 13 segregating $F_2$ mapping populations by intercrossing 20 *flc* KO mutants, representing 16 accessions. The mutant pairs, from genetically distinct backgrounds, were chosen to provide contrasts in flowering time (overall phenotypic range of DTF = 14.1 to 38.0 and RLN = 4.8 to 32.6) (S5B Fig). All together, they represent the full phenotypic flowering time spectrum of our collection of *flc* KO mutants. We measured flowering traits in these $F_2$ populations (Figs 2A, S5 and S6 and S3 Table), finding limited transgressive variation, especially on the early side (Fig 2B), in contrast to $F_2$ populations with segregating functional *FLC* alleles [22]. The extent to which flowering traits were correlated with each other varied greatly in the $F_2$ populations (Figs 2C and S7). This was different from what was seen in the parents of the crosses (Fig 1E), indicating that trait canalization due to *FLC* disruption became uncoupled in specific recombinant backgrounds.

Using an improved TIGER method for the detection of recombination breakpoints [30], we identified a total of 115 additive QTL across all $F_2$ populations, with an average of 2.25 QTL per population × phenotype combination (S8A and S4 Figs and S5 Tables) and with explained additive variation ranging from 10% to 66% (S8B Fig). For 99 out of 115 QTL, the early parent contributed all flowering-promoting alleles (S8C Fig). The combined QTL effects predicted the differences in flowering time between parents well (simple linear model, multiple $R^2_{adj} = 0.68$, $p = 0.0005$) (S8D and S8E Fig). Two-dimensional genome scans revealed only three spurious, minor-effect QTL interactions, in stark contrast to crosses with fully functional *FLC* alleles [18,22–26,31], indicating that additive flowering-time QTL become prevalent once *FLC* is inactivated (S5 Table). This is in agreement with what has been observed when flowering-time QTL were mapped in populations that were allowed to overwinter outdoors, conditions under which neither *FLC* nor *FRI* make major contributions

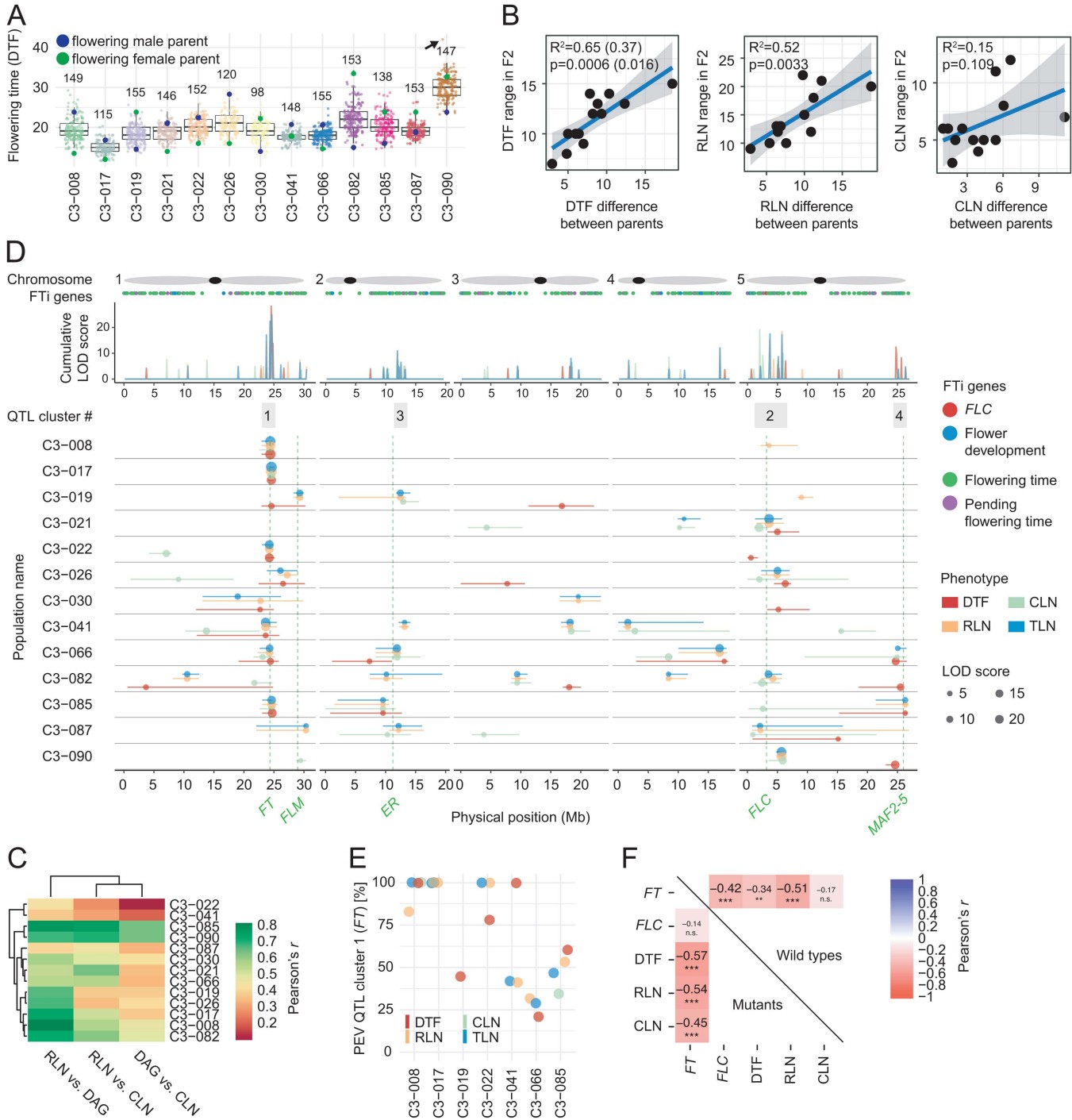

**Fig 2. Quantitative genetic analysis of *FLC*-independent flowering. (A)** Distribution of flowering time (DTF) of F₂ individuals, including the mean value of the respective *flc* mutant parent lines. The numbers of F₂ individuals per population are shown on top. The arrow indicates a single outlier of population C3-090, which was excluded in the correlation shown in **C**. **(B)** Correlation between the phenotypic range of the parents and of the F₂ populations (range = F₂(max) − F₂(min)). Simple linear model, DTF, excluding a single outlier of population C3-090 (see panel **A**): $r^2_{adj}$ = 0.65, *p* = 0.0006; including the outlier: $r^2_{adj}$ = 0.37, *p* = 0.016; RLN, $r^2_{adj}$ = 0.52, *p* = 0.0033; CLN, $r^2_{adj}$ = 0.15, *p* = 0.109. **(C)** Flowering trait correlations in F₂ populations. Range of Pearson's *r*: RLN versus DTF 0.41 to 0.85; RLN versus CLN 0.31 to 0.80; DTF versus CLN 0.13 to 0.68; all *p* < 0.0001. **(D)** Summary of QTL analysis. On top, schematic representation of chromosomes, with black dots representing centromeres. The physical position of genes with a known role

in flowering is shown below (S6 Table), with color indicating the published classification [33,34]. LOD scores were summed over a non-overlapping moving window of size 100 kb and are shown at the center of the window. The detected QTL clusters 1–4 are indicated on top, the widths of the gray boxes indicate the size of each cluster. Cluster 1 is a 2 Mb region (±1 Mb from QTL LOD peak at 24.675 Mb). Cluster 2 is a 7 Mb region (0–7 Mb) on chromosome 5. Clusters 3 and 4 are smaller and contain fewer QTL. The QTL intervals (95% Bayes interval) are shown as horizontal lines, and the physical positions of *a priori* flowering candidate genes are indicated as dashed vertical green lines. **(E)** Proportional explained additive variation (PEV) of QTL co-localizing with *FT*. **(F)** Correlation of *FT* expression levels with flowering traits. Mutants on the lower and wild types on the upper triangle. Pearson's *r*, mutants, −0.57 to −0.45, $p < 0.001$, 109 *d.f.*; wild types, DTF: $r = -0.34$, $p < 0.01$, RLN $r = -0.51$, $p < 0.001$, 59 *d.f.*. The significance of the correlations is indicated by *, $p \leq 0.05$; **, $p \leq 0.01$; ***, $p \leq 0.001$; n.s., not significant. The data underlying this figure can be found in S1 Data and https://doi.org/10.5281/zenodo.15403194.

to flowering time [32]. Across all $F_2$ populations, two major and two minor QTL clusters were apparent, with QTL affecting only one of the three flowering traits being the exception (Figs 2C, 2D and S8C and S5 Table). We conclude that while genetic coupling of flowering traits is common, some traits can be unlinked in specific genetic backgrounds.

The major QTL cluster 2 on top of chromosome 5 co-localizes with *FLC* (Fig 2D and 2E). Flowering time QTL near *FLC* have been identified before, but the strong effects of *FLC* itself have masked the contribution of other genes in this interval [22,23,31]. As the use of *flc* KO alleles in all parents excludes *FLC* as causal, we conclude that the top of chromosome 5 is a general hotspot for vernalization-dependent and -independent flowering time variation (Fig 2D and S5 Table). Given the width of the cluster 2 around *FLC*, which includes *CONSTANS* (*CO*) and 24 other known flowering time regulators (S5 Table), it appears that the strong effects of *FLC* obscured in many earlier QTL studies the effects of allelic variation at several other loci in this region.

Finally, major cluster 1, which was defined by several QTL in a narrow region near the bottom of chromosome 1 (Fig 2D and 2E), explained all of the additive variation (PEV) in three $F_2$ populations with one extremely early parent, and some variation in several other $F_2$ populations. A strong candidate for cluster 1 is the FLC target *FT*, for which extensive functional variation in the non-coding region has been documented [32,35–39]. Compared to previous studies, however, cluster 1 explained significantly more variation as a single QTL in our study. As observed before [40,41], relative *FT* transcript levels correlated well with flowering traits in both *flc* mutants and wild types (Figs 2F and S4), and its expression was significantly upregulated in many *flc* mutants (S3 Fig). We conclude that release of *FLC* repression magnifies the effects of differences in *FT* activity.

## Extremely early-flowering individuals in Southern Italy

Since a reduction of *FLC* flowering-repressing activity, either through *FRI* or *FLC* mutations, is common in natural populations of *A. thaliana*, and since alleles with strongly flowering-promoting *FT* alleles are common as well, we were wondering why there were no reports in the literature of natural accessions that flower as early as our artificial material does in controlled conditions (S9A Fig). Had we created combinations of *FLC* and *FT* alleles that were deleterious in natural conditions? Or had extremely early flowering individuals in nature been missed by previous collection efforts? We integrated a phylogenetic analysis of *FT* sequences from 1,135 accessions with data on *FLC* expression and flowering time as indicators of the life history strategy exemplified by different accessions [27,39,42]. The *FT* alleles of the earlier parents of the five $F_2$ populations with a QTL at *FT* fell into several different phylogenetic clades, which were not restricted to late-flowering, winter-annual accessions (Fig 3A). We therefore hypothesized that extremely early flowering accessions do exist in nature and that they would most likely have evolved where rapid flowering is an escape strategy under drought, for example, around the Mediterranean [43]. Since we had cursorily noticed very early flowering populations on previous collecting trips, we prospected for early-flowering plants among two Southern Italian populations (Figs 3B, 3C and S9B–S9K). In the greenhouse, the progeny from one of the locations, Angit, included seven individuals, collected over three years, that flowered as early as our earliest *flc* mutants (Fig 3D). Given the high plasticity of flowering traits in *A. thaliana*, it was not surprising that the field observations did not predict the phenotypes in controlled conditions well (simple lm, adj. $R^2$: 0.155, $p > 0.05$). Nevertheless, five of 16 accessions flowered similarly early in both environments (S9L Fig).

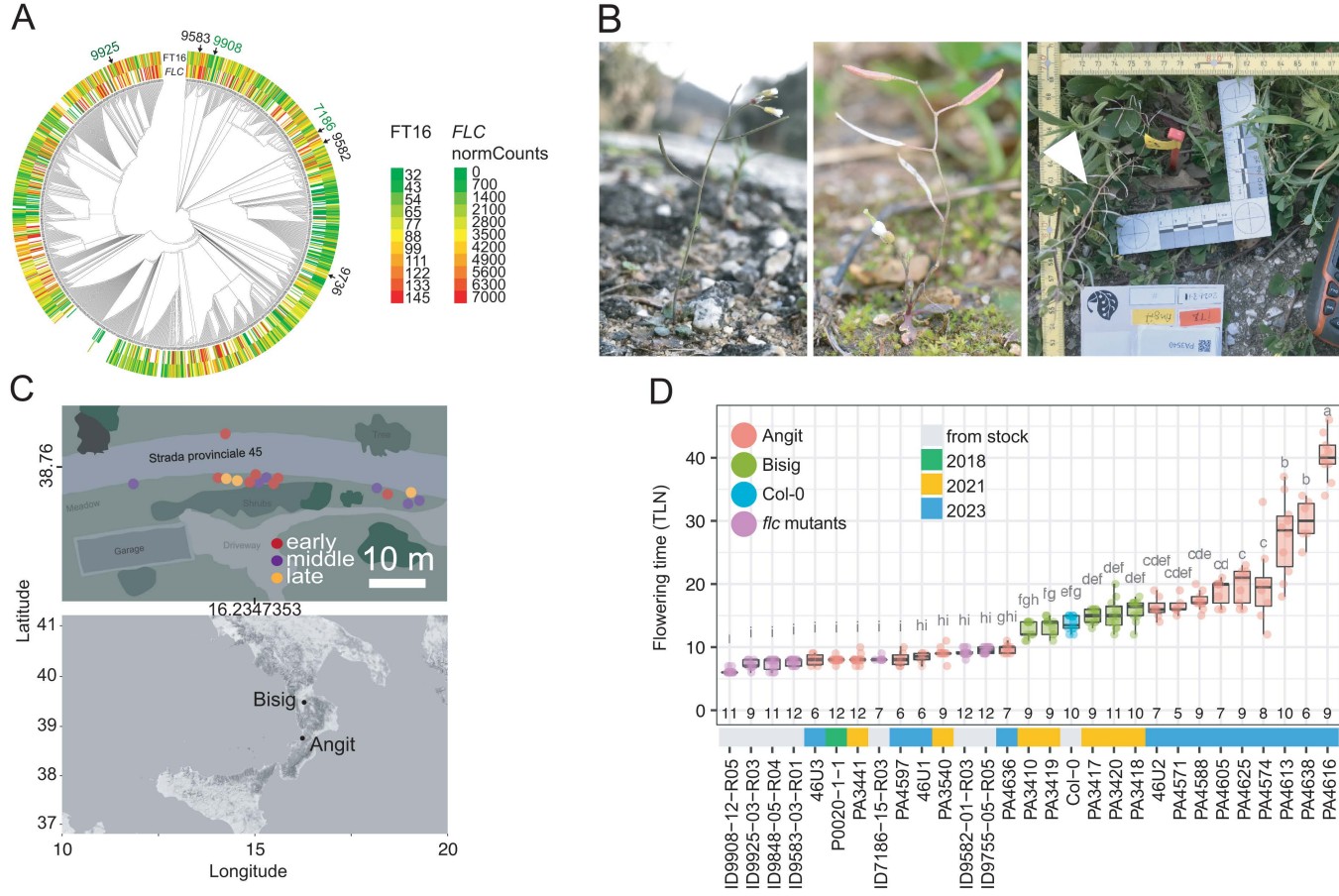

**Fig 3. Discovery of very early flowering genotypes in Italy.** (A) Comparison of *FT* phylogeny with flowering time at 16°C (FT16) [39] and *FLC* expression (*FLC* normCounts) [42]. The earliest flowering parents of the five F$_2$ populations with a QTL at *FT* are indicated with an arrow. (B) Early-flowering plants in the wild (at the Angit site). (C) Southern Italian sampling sites. Schematic map on top shows the Angit site. Colors roughly indicate apparent flowering times. (D) Flowering time of progeny from wild plants and control genotypes in the greenhouse (22°C long days). "IDXXXX" is the wild-type strain ID from the 1001 Genomes Project (https://1001genomes.org). The subsequent digits are from a consecutive numbering system for lines selected in the T2 and T3 generations. Similar letters indicate no significant difference in total leaf number (RLN + CLN) (ANOVA with post hoc Tukey HSD, *p*.adj. < 0.05). The data underlying this figure can be found in S1 Data and https://doi.org/10.5281/zenodo.15403194.

Taken together, we found that new mutant phenotypes created in the laboratory may point to portions of the natural phenotypic spectrum of wild species that have apparently been missed because these phenotypes were simply not expected by collectors; in this case, extremely early flowering. We conclude furthermore that extremely early flowering in specific genetic backgrounds can apparently be achieved by changes at only two loci, *FT* and *FLC*.

## Variation in pleiotropic roles of *FLC*

*FLC* regulates not only flowering, but also the switch from the juvenile to the adult vegetative phase, with flowering-independent effects on leaf shape [44]. Even broader roles of *FLC* beyond life-history transitions can be inferred from the binding of FLC protein to promoters of hundreds of target genes with many different functions [9,10]. To begin to examine the extent of background-dependent effects of *FLC*, we looked at growth rate. Growth, biomass accumulation, and leaf structure are ecophysiologically relevant traits that often vary in a coordinated manner, forming a trade-off known as the leaf economics spectrum (LES) [45,46]. The LES is tightly linked to the so-called slow-fast-continuum, which is prevalent

PLOS Biology

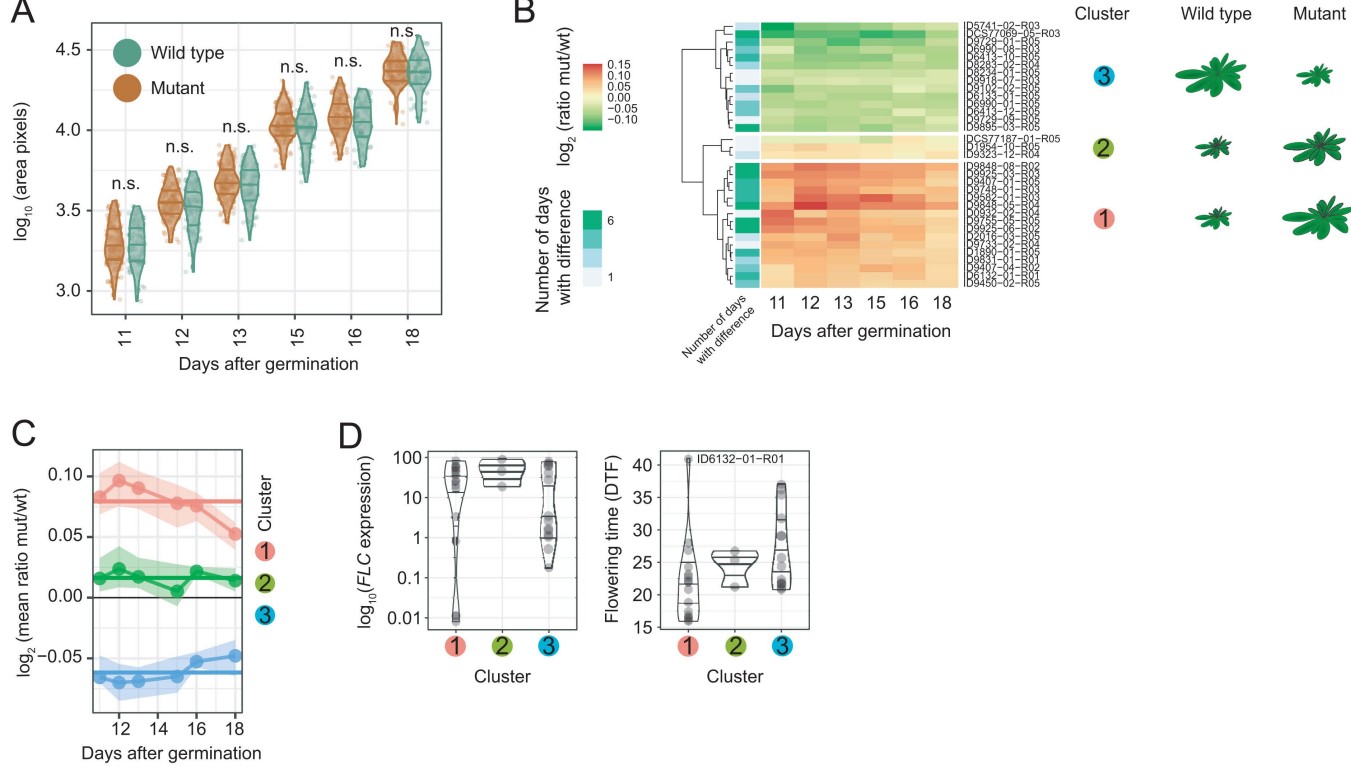

**Fig 4. Growth trajectories of *flc* mutants. (A)** Projected rosette area (PRA) of 61 wild types and 112 mutants per day (*x*-axis). PRA of the 8376 wild type could not be determined. Differences between mutants and wild types on each day were not significant (n.s.) (Mann–Whitney *U* rank test, Bonferroni correction, *p*.adj. > 0.05). **(B)** Clustering by PRA (log₂) of 110 mutants versus wild types. The number of days with significantly different projected rosette area measurements (Mann–Whitney-*U*-test, Benjamini–Hochberg correction, *p*.adj. > 0.05) is indicated. A schematic representation shows the cluster size differences. **(C)** Cluster mean values of the ratio on each day (dots) and over all days (horizontal lines, log₂). The values between 25th and 75th percentiles are shown as ribbons. **(D)** FLC levels (log₁₀) and flowering time of mutants in cluster 1–3. The data underlying this figure can be found in S1 Data and https://doi.org/10.5281/zenodo.15403194.

in many plants [47] and refers to the co-occurrence of either earlier flowering and faster growth, or later flowering and slower growth.

Comparing the relative growth rates between the *flc* mutants and the corresponding wild types, we did not find major differences (S10A and S10B Fig). To test how *FLC* effects on plant size depend on the genetic background, we measured the projected rosette area (PRA) as a proxy for biomass and thus fitness, before bolting slows down growth. While the sizes of mutants and wild types were overall similar (Fig 4A), 33 mutants differed from the corresponding wild types on at least one day (Fig 4B).These fell into three major clusters according to the trajectory of size differences. The most interesting, cluster 3, included 14 *flc* mutants representing 11 accessions that were consistently smaller than the corresponding wild types (Mann–Whitney *U* rank test, *p*.adj. < 0.05) (Figs 4B, 4C and S10C). No differences in flowering time (DTF) and *FLC* transcript levels were observed between the mutants of the three clusters (Kruskal–Wallis *H*-test, *p* > 0.05) (Fig 4D), but all the earliest mutants (DTF < 20) were in cluster 1 and thus larger on all days, as expected for an early flowering/fast growth strategy.

To reveal potential background-specific regulatory roles of *FLC*, we selected seven *flc* mutants that both flowered early (DTF < 30) and had low *FLC* transcript levels (*FLC* RT-qPCR level range 0.1 to 11.7 a.u., all KOs, except ID9402-01-R05 = 11.7 a.u.) for RNA-seq analysis. We identified 1–17 differentially expressed genes (DEGs), for a total of 31

DEGs, which included the flowering regulators *SOC1* and *SVP*. Of the remaining DEGs, 27 were restricted to one mutant-wild type contrast, and one DEG, AT5G22580, which is not a known FLC target, was shared between two contrasts. The known target *PAP16* was strongly upregulated in one *flc* mutant ([Fig 5A]). Due to the timing of the sampling during the transition to flowering we were not able to distinguish between direct effects of *FLC* and effects that are indirect consequences of a change in flowering time in *flc* mutants. Nevertheless, the magnitude of the flowering-time differences was not predictive of DEG sets, indicating that regulatory effects of *FLC* and the genetic networks in which *FLC* participates

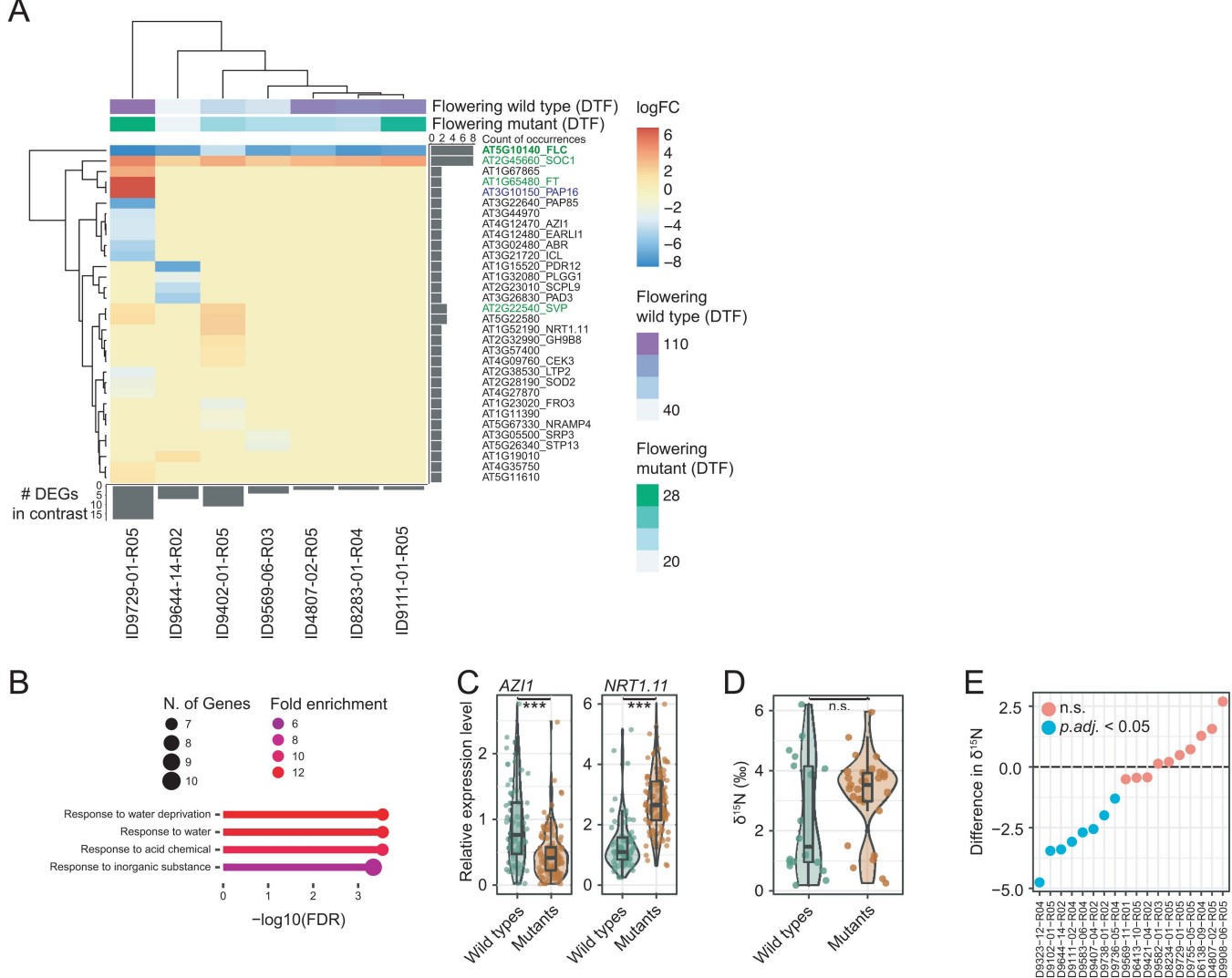

**Fig 5. Evidence for pleiotropic roles of *FLC*. (A)** Heatmap of the $\log_2 FC$ values of differentially expressed genes (DEGs) present in at least one contrast. Flowering time data (DTF) from the experiment shown in [Fig 1A] and [1B]. **(B)** GO enrichment analysis. The top 5 hits (−log(FDR)) are shown with an FDR cutoff of 0.01. **(C)** Relative expression levels [a.u.] of *AZI1* and *NRT1.11* in all the wild types and the mutants. *AZI1*, mean wild types: 0.92 a.u., mean mutants: 0.47 a.u., *NRT1.11*, mean wild types: 0.47 a.u., mean mutants: 1.37 a.u. **(D)** Nitrogen isotope composition ($\delta^{15}N$ [‰]) in a subset of 29 early flowering mutants and the corresponding wild types (means of three biological replicates). Mean [±se] mutants: 3.17‰ [±0.26]; wild types: 2.43‰ [± 0.46]; Mann–Whitney *U* rank test, $p > 0.05$. **(E)** Difference in $\delta^{15}N$ ($\delta^{15}N_{diff} = \delta^{15}N_{Wildtype} − \delta^{15}N_{Mutant}$) between wild type and the respective mutant. Turquoise indicates significant differences, two-sided Student's *t* test, Benjamini–Hochberg correction, *p*.adj. < 0.05. The data underlying this figure can be found in [S1 Data] and https://doi.org/10.5281/zenodo.15403194.

vary between genetic backgrounds. While having only 31 DEGs limits the power of GO enrichment analysis, one of the enriched categories was related to water stress ($-\log_{10}$(FDR) > 2) (Fig 5B), consistent with a role of the *FRI/FLC* module in regulating water use efficiency [48,49], which becomes uncoupled from flowering time in specific *flc* mutants [14]. Additional pleiotropic roles of *FLC* can be inferred from altered expression of *AZI1* and *EARLI1*, two genes with roles in systemic defense priming and control of root-growth under Zn-limiting conditions [50,51] (Fig 5C), in two *flc* mutants.

Examining *AZI1* expression in our entire collection of *flc* mutants revealed that *AZI1* expression was on average lower across all *flc* mutants (Mann–Whitney-*U*-test, $p = 9.4e{-}08$) (Fig 5C). While the differences in *AZI1* levels in individual pairwise comparisons were not robust to correction for multiple comparison (two-sided Student's *t* test, Benjamini–Hochberg correction, *p*.adj. > 0.05), 12 mutants (representing seven wild types) had reduced *AZI1* levels (two-sided Student's *t* test, $p < 0.05$) (S11A Fig).

Nitrate can affect flowering partially via *FLC* [52,53]. Because RNA-seq data had indicated that the nitrate transporter gene *NRT1.11*, which is required for the transfer of root-derived nitrate into phloem in the major veins of mature leaves [54,55], was upregulated in one *flc* mutant ($\log_2$FC [FDR]: 2.5 [0.025]) (Fig 5A), we also measured *NRT1.11* RNA levels in our entire collection. Across all *flc* mutants, *NRT1.11* levels were increased (Mann–Whitney-*U*-test, $p = 2.2e{-}16$) (Fig 5C). Before multiple testing correction, 33 individual mutants representing 28 wild types showed differential expression of *NRT1.11*, which was upregulated in all but one case (two-sided Student's *t* test, Benjamini–Hochberg correction, *p*.adj. < 0.05) (S11B Fig). In one *flc* mutant, the difference remained significant after correction for multiple testing.

Under non-limiting nitrate conditions, nitrogen isotope ratios ($\delta^{15}$N [‰]) can serve as proxies for *in planta* nitrogen use efficiency [56]. We measured $\delta^{15}$N in 29 early flowering *flc* mutants from 19 accessions (with a maximum relative *FLC* expression level of 18.3 a.u.) and the corresponding wild types. Only the wild types had a clearly bimodal $\delta^{15}$N distribution. While the $\delta^{15}$N levels in *flc* mutants and wild types were overall similar (mean [±se] mutants: 3.17‰ [±0.26], wild types: 2.43‰ [±0.46], Mann–Whitney *U* rank test, $p > 0.05$), pairwise comparisons of 18 *flc* mutant/wild type contrasts revealed eight mutants with higher $\delta^{15}$N (two-sided Student's *t* test, Benjamini–Hochberg correction, *p*.adj. < 0.05) (Figs 5D, 5E and S12A). $\delta^{15}$N correlated neither with flowering time nor *FLC* expression nor $\delta^{13}$C levels (see ref. [14]). The carbon-to-nitrogen (C/N) ratio in plant tissue, which is directly linked to photosynthetic activity, was also unchanged in the *flc* mutants (two-sided Student's *t* test, *p*.adj. > 0.05) (S12B Fig) [57,58]. Together, this suggests that *FLC* has background-specific pleiotropic effects on $\delta^{15}$N that are independent of *FLC* effects on the other investigated traits.

## Discussion

Genetic analyses in model organisms are typically restricted to one or a few laboratory strains, often chosen because of their ease of cultivation or husbandry. This is also true for *A. thaliana*, even though it has been known for decades that mutant effects can sometimes differ dramatically between genetic backgrounds [59,60]. Here, we leveraged a new species-wide *flc* mutant collection to uncover a previously overlooked part of the flowering time spectrum in natural accessions. In addition, we report background-specific roles of *FLC* in other processes, further underscoring the value of a pan-genetic approach [13].

We used cost-efficient quantitative genetics at scale to determine the *FLC*-independent genetic architecture of flowering traits by crossing *flc* KO lines in genetically diverse backgrounds and with contrasting phenotypes. One of the QTL clusters we identified, on the bottom of chromosome 5, overlaps with the location of *FLC* homologs from the *MAF* family, which have been implicated in vernalization response [61], and these might explain the remaining variation in vernalization sensitivity in *flc* mutants. Another major-effect QTL co-localizes with the central flowering regulator *FT*, explaining extremely early flowering in several *flc* mutants. The proportion of explained variation for this QTL is much greater than what has been typically observed before [18–29], reflecting the special nature of our material.

Our observations of extremely early flowering led us to prospect for extremely early-flowering plants in nature, which apparently had been overlooked by previous collectors. The underrepresentation of extremely early flowering accessions

in *A. thaliana* seed collections is likely due to the fact that such individuals are ephemeral and that they can be found only during a very short time window. Given the relatively small number of accessions and growth conditions we examined here, we suspect that the lower limit of flowering in natural accessions has not yet been found. A possible ecological niche for extremely early flowering could be an environment that supports only a very short life span, due to lethal conditions such as terminal drought. Early flowering, a common drought escape strategy, would then manifest in an extreme way in "super escapees", which could have advantages in regions like Southern Scandinavia where the growth season was predicted to be too short to allow for drought escape by "normal" summer-annual, early flowering accessions [62,63]. Given the ample genetic opportunities *A. thaliana* has to accelerate flowering and to escape drought, taken together with potential sampling bias, it seems likely that we have engineered a phenocopy of accessions that are part of the phenotypic spectrum that either exists already today or that will become highly favorable with a changing climate in many regions.

Contrasting flowering behavior is indicative of specific trait syndromes and co-variation with biomass accumulation is common [12,64]. In our study, many *flc* mutants were larger during the early phase of vegetative growth, suggesting that inactivation of *FLC* pushed many accessions towards an early flowering/fast growth strategy [46,64]. However, there were also *flc* mutants that grew more slowly, which is opposite to what would be expected from the prevailing slow-fast-continuum. One explanation could be the presence of harsh environmental conditions at the origin of these accessions. Because slow growth increases drought tolerance, the combination with early flowering growth could be particularly advantageous in conditions with severe conditions of drought [65,66]. Staying small could also be beneficial under conditions of high pathogenic pressure, since expression of immunity genes is linked to flowering in natural populations of *A. thaliana* [67]. Similarly relevant to pleiotropy, all flowering traits became highly canalized in *flc* mutants, similar to what we saw when *FLC* activity was reduced by vernalization.

Despite the limited scope of our gene expression analyses, the results also support variation in the pleiotropic roles of *FLC*. Neither absolute flowering time nor the relative acceleration of flowering in *flc* mutants explained the main differences in differentially expressed genes patterns, suggesting that most effects were related to the specific genetic backgrounds. Since the early flowering *flc* lines that were examined by RNA-seq clustered in different groups in the growth analysis, they did not provide further insights into functions of *FLC*-independent regulators on pleiotropic phenotypes. Orthologous analysis of FLC binding sites in the genomes of *A. thaliana* and its relative *Arabis alpina* have revealed that fewer than one in five are conserved, and many of the nonconserved target genes are involved in stress responses [11]. Likewise, we find that stress-related genes are overrepresented among DEGs that are specific to subsets of accessions or even individual accessions. Moving forward, it will be of interest to study gene expression changes in non-inductive short days.

In conclusion, we have demonstrated how the genetic modification of one trait that is part of a complex trait syndrome can unmask deviations from general trait associations. These "outlier situations" are particularly interesting candidates for functional follow-up studies, as they might provide explanations on how associations are restructured to adapt to niche environments. Background-dependent pleiotropic effects are also a major limitation in breeding, as recently highlighted by editing orthologs and paralogs in multiple Solanaceae, an approach that the authors termed "pan-genetics" [13]. In the same vein, our collection of species-wide mutations in the same gene, presented before [68] and in this study, provides an excellent avenue for understanding the true extent of genetic networks in which the focal gene participates. Such pan-genetic approaches could generally help to engineer quantitative trait variation that goes beyond what is observed in the original population, even if it already contains substantial functional variation of a gene of interest.

## Materials and methods

### Genetic resources

*flc* **mutants.** The generation of our collection of *flc* mutants has been described in ref. [14].

**Generation of $F_2$ mapping populations.** KO mutant lines covering most of the flowering time range in the mutant population were crossed. Around five $F_1$ plants of each cross were selfed to obtain biparental $F_2$ mapping populations,

from which one was randomly selected for the mapping experiment. The identity of the parents of the mutant lines and of the $F_1$ individuals was verified with shallow Illumina short-read whole-genome sequencing (WGS) and SNPmatch (version 5.0.1) [69].

**Plant collections in Southern Italy.** Plant populations were identified in 2018 as part of the Pathodopsis collection [70], and based on their proximity to the coordinates listed as original collection sites of accessions from the 1001 Genomes Project (https://1001genomes.org): 38.76°N, 16.24°E (Angit) and 39.48°N, 16.28°E (Bisig). In March 2018, a single Angit plant (P0020-1) that was already mature at the time of visit was collected in a seed bag. In 2021 and 2023, plant collections in March and April followed a linear transect through the entire extent of both populations, and roughly every 20th plant encountered was selected for seed propagation. The precise coordinates of the plants harvested for this study are listed in S7 Table. Prior to harvesting, plants were photographed with anApple iPad Pro, usually including a photomacrographic scale ABFO No.2 (cop-shop.de) as size reference. Collection spots were either directly geo-referenced with a GPS (Garmin GPSmag 64S) or indirectly by distance to a geo-referenced point along the transect. Mature plants were then directly transferred into a seed bag, less mature plants were transferred, with their roots, into pots with potting soil (CL P, einheitserde.de). After transport back to Germany by car, plants in pots were grown on a windowsill until seeds could be harvested. From every seed stock obtained in 2018 and 2021, a single individual was propagated in a growth chamber to obtain a fresh seed stock. In 2023, seeds from field plants were directly used for experiments after an after-ripening period of at least two months.

## Phenotyping and growth conditions

**Experimental design.** Unless described otherwise, experiments for phenotyping were conducted with 12 biological replicates per line. Four groups of three biological replicates were randomly assigned to one tray with 60 pots. The position of each group of three biological replicates within a tray was assigned randomly. In every phenotyping experiment, all trays were rotated 180º and moved every other day to minimize position effects. No block effects were detected for the measured traits (One-way ANOVA, Bonferroni corrected, $p > 0.05$).

**Greenhouse growth conditions.** Plants were grown under constant temperature of 22°C under long-day (LD) conditions (16 h light/8 h dark) and 65% humidity. Natural light was supplemented with LED arrays (Valoya, Model BX180c2, Spectrum AP67) to reach 120–150 µmol m$^{-2}$ s$^{-1}$ photosynthetic photon flux density. Seeds suspended in water were stratified for five days in the dark at 4°C. Around five to 10 seeds were sown in each pot with ED73 potting mix (Einheitserdewerke, Sinntal-Altengronau, Germany). At the full expansion of the first two true leaves, plants were thinned to retain only one plant per pot. Because of reduced germination in some pots or severe damage of plants during thinning, the number of replicates per line varied.

**Flowering time analysis.** Flowering time was assessed through rosette leaf number (RLN), cauline leaf number (CLN) on the main shoot, and days to appearance of the floral bud (DTF). RLN and CLN were individually determined and combined to obtain total leaf number (TLN). DTF was consistently recorded, with a maximum one-day gap, representing the duration from germination to the emergence of the floral bud. Instances where flowering occurred later than 125 days or not at all were categorized as DTF 130, including the corresponding rosette leaf count at that time. Throughout the experiment, water status remained constant. In the experiment depicted in Fig 1A and beyond, low germination rates hindered the analysis of one wild type (8,376), and RLN and CLN could not be conclusively measured for seven additional wild types, as they did not flower during our experiment. A DTF value of 130 was assigned to these eight wild types.

**Vernalization treatment for flowering time analysis.** Seeds were germinated at 22°C under LD and transferred to 4°C under short days (SD, 8 h light/16 h dark) when cotyledons began to expand. After 60 days, the trays were transferred to the greenhouse with conditions described before. The vernalization period was subtracted from the DTF measurements.

**Measurement of plant growth.** Plant growth was monitored daily by capturing top-view pictures using an EOS 2000D digital camera (Canon). Trays were identified by inclusion of triple-redundant QR codes. Images were normalized for size, orientation and perspective. This required between 15 and 45 s per image, where a quadrangle had to be placed at predetermined marker positions to compute a transformation matrix. A web-based tool was used for the interactive part (Labelbox, https://labelbox.com). Following normalization, a segmentation was performed, where the background was removed and the individual plants were extracted from the normalized multi-plant tray images. Background removal was performed by first applying a threshold on the images in the 'Lab' color space, followed by a series of morphological operations to remove noise and non-plant objects and a GrabCut-based postprocessing. The workflow was implemented in Python 3.6 and bash using OpenCV 3.1.0 and scikit-image 0.13.0 for image processing. Post-experiment, it was observed that the supplementary LED light in the greenhouse adversely affected image analysis. Consequently, images captured on days with the active supplemental lighting were excluded from the analysis. Moreover, plants with sizes below 5,000 pixels in the later stage of measurements and those smaller than 500 pixels on any day within the specified time frame were entirely omitted from the dataset. These exclusions were made to eliminate likely empty pots or severely affected, dying plants (e.g., during thinning). Subsequently, area values in pixels underwent log transformation for subsequent analysis.

## Quantitative genetic analysis

**Experimental design.** $F_2$ plants were cultivated in trays containing 60 pots, with four trays assigned to each $F_2$ mapping population. Six plants from each parent line were randomly placed across the four trays. While all trays for a mapping population were kept together in the greenhouse, they underwent regular rotation, and the entire group of trays was relocated every second day to minimize positional effects. Growth and flowering time were analyzed following the procedures outlined in the "Flowering time analysis" section. Due to insufficient germination in certain pots, the $F_2$ plant count per population varied.

**DNA extraction.** DNA was extracted according to ref. [71] with Econospin 96-well filter plates (Epoch Life Science, USA). The extraction buffer was modified to contain MES sodium salt instead of MES hydrate and RNase A (QIAGEN GmbH, Hilden, Germany).

**WGS shallow sequencing.** After DNA extraction and WGS library preparation, 150 bp paired-end reads were obtained on a HiSeq3000 instrument (Illumina, San Diego, USA). The mean [±sd] number of reads of all $F_2$ samples was 890,202 [356,734], ranging from 1,672 to 2,187,790.

**Marker generation.** To extract informative genetic markers, we enhanced the Trained Individual Genome Reconstruction (TIGER) CO analysis pipeline [30] to be compatible with mapping populations of non-Col-0 (TAIR10) parents. The refined pipeline features an automated variant filtering step and a streamlined Snakemake-based process that outputs marker data compatible with the popular mapping package R/qtl [30,72]. The Snakemake pipeline, requiring minimal user input, generates cross-type marker input files suitable for rRqtl (https://github.com/ibebio/tiger-pipeline and https://doi.org/10.5281/zenodo.15535778).

Preprocessing involved trimming reads, mapping them to the TAIR10 reference, and removing duplicates. Variant calling for parent samples was performed with GATK, with the parent providing a higher number of variants selected as the source for alternative variant information (ALT). A "complete" variant file was obtained after soft filtering with GATK VariantFiltration ("QD < 5.0 || FS > 60.0 || MQ < 50.0 || MQRankSum < −12.5 || ReadPosRankSum < −8.0") and extraction of biallelic SNPs. Subsequently, a "corrected" variant file was obtained through automated filtering, involving removal of variants deviating from the dominant peak of a bimodal Gaussian distribution fitted to QD values by SD*2.5, unimodal Gaussian distributions of FS by SD*2.5, of MQRankSum by SD*4 and MQ fixed at 50. Variants in centromeres, telomeres, and regions with transposable element annotations were excluded.

The complete marker file served as a reference for variant calling of $F_2$ samples with GATK CollectAllelicCounts, excluding variants in organellar genomes and those with coverage deviating by five times the standard deviation of the mean. Samples with fewer than 7,000 variants were removed. Users can swiftly adjust these parameters through a configuration file. The prepared population-specific complete and corrected marker files, along with $F_2$ allele count files, were employed as input for TIGER analysis, with slight script modifications to ensure file compatibility. The TIGER output files (one per $F_2$ sample) were merged into a single R/qtl cross-type input file per population, and QC reports and genotype plots were generated at multiple steps during the pipeline.

**QTL mapping.** Phenotypic data was integrated into each R/qtl input file, and markers were filtered using R/qtl [73]. $F_2$ individuals with a low number of called markers, markers with substantial missing information, and individuals with very similar genotypes (90% similarity cutoff) were excluded. Markers with strong segregation distortion ($p < 1e{-}7$) and individuals with over 25 crossing-over events (more than twice than the expected median) were also removed. TIGER-identified crossing-over events served as unique markers in each $F_2$ population, ensuring consistent genetic distances between markers. These prepared marker files were utilized for QTL mapping with a slightly modified version of the foxy QTL pipeline, which employs the R/qtl package in R [74] (https://github.com/maxjfeldman/foxy_qtl_pipeline). Single-QTL model genome scans utilized Haley–Knott regression to identify QTL with LOD scores exceeding the significant threshold, determined through 1,000 permutations at alpha = 0.05. For 2D genome scans, we employed a two-QTL model, with the significant threshold determined through 100 permutations at alpha = 0.05. A stepwise forward/backward selection procedure was then conducted to identify an additive, multiple QTL model based on maximizing the penalized LOD score [72,74]. Following the concatenation of all QTL tables for all populations and traits, physical positions were assigned to each QTL, along with their respective 95% Bayes intervals.

## RT-qPCR expression analysis

Performed as described in ref. [14]. All primer sequences are listed in S8 Table.

## RNA-seq

After normalization, the same RNA as used for the RT-qPCR experiment was used for library preparation using an in-house custom protocol adapted from Illumina's TruSeq library prep, with details provided in ref. [75]. FASTQ files from multiple lanes were merged and mapped to the TAIR10 transcriptome using RSEM (bowtie2, version 2.2.3) with default parameters. A mRNA counts file was obtained with feature counts (version 1.6.1). One of three biological replicates of line ID9402-01-R05 (replicate 3) was removed due to an insufficient number of feature counts (690,569). Differential expression analysis was conducted with edgeR, with model.matrix (approximately 0 + group) as design and the function makeContrasts and glmTreat to retrieve mutant-wild type contrast specific lists of DEGs (FDR < 0.1 and $|\log_2$ FoldChange$| > 1$). GO enrichment analysis was conducted with ShinyGO with the pathway database "GO Biological Process" with 22,157 gene IDs as background and a FDR cutoff of 0.01 [76].

## *FT* and *FLC* sequence analysis

SNP variants at *FT* from the 1001 Genomes resource (http://1001genomes.org/data/GMI-MPI/releases/v3.1/) were extracted using VCFtools (version 0.1.16) with "--chr 1 --from-bp 24325373 --to-bp 24335992". VCF files were transformed to fasta using PGDSpider (version 2.1.1.5) and aligned with muscle (version 3.8.31). A neighbor-joining tree was built with MEGA X [77]. The dataset comprised 1,135 sequences. All positions with fewer than 95% coverage were eliminated, resulting in 382 positions. A tree was visualized with iTOL [78]. Variants at *FLC* were extracted using VCFtools "--chr 5 --from-bp 3173382 --to-bp 3179448" and filtered for SNPs with a minor allele frequency of 10% and maximum missing data of 10% with "--remove-indels --maf 0.1 --max-missing 0.9", resulting in 35 variants. Missing genotypes were imputed

with Beagle 4.0 (version r1399) and a Minimum Spanning Network was constructed with PopArt [79]. The *FLC* expression data were from ref. [42] and flowering time data from https://arapheno.1001genomes.org/phenotype/262/.

**Measurement of nitrogen isotope ratio and carbon and nitrogen content**

Mutants and wild types were grown in three pots representing three biological replicates grown at 22°C under LD in the greenhouse. The pots were randomly distributed over a total of 10 trays, which were rotated and moved every day to reduce position effects. After allowing germination and establishment of the first true leaves, plants were thinned to three plants per pot. The leaves of different plants never overlapped with each other during the experiment. Rosettes were harvested at the initiation of flowering or after 22 days after germination (whatever was first). Depending on the rosette size at the initiation of flowering several plants were combined to one replicate to reach the required amount of tissue for analysis. Plant tissue was dried at 60°C for 24 h and homogenized in 5 ml tubes (Eppendorf, Germany) containing 5 ball bearings to very fine and uniform powder. Dried material was transferred to a 1.5 ml microfuge tube and sent to Isolab (Schweitenkirchen, Germany) for an analysis of nitrogen isotope composition and C and N content ($\delta^{15}N$, %N and %C) with $^{15}N$-CF-IRMS and $^{13}C$-CF-IRMS. Four technical replicates per sample were analyzed. For a more detailed description of the procedure, see ref. [80]. Data are presented as $\delta^{15}N$ [‰] versus AIR, and percent mass carbon (%C) and nitrogen (%N) in the plant tissue, respectively.

## Supporting information

**S1 Fig.** Flowering time data of wild types and mutants, arranged by wild type. Mean values are shown in Fig 1A and 1B. The data underlying this figure can be found in S1 Data and https://doi.org/10.5281/zenodo.15403194. (EPS)

**S2 Fig. Pearson's correlation analysis of *FLC* transcript levels and the flowering traits.** DTF, RLN and CLN (untransformed mean values) with and without vernalization in mutants **(A)** and wild types **(B)** are shown, with *p*-values in the upper triangles. Summarized data shown in Fig 1E. The data underlying this figure can be found in S1 Data and https://doi.org/10.5281/zenodo.15403194. (EPS)

**S3 Fig. *FT* expression in nine-day-old plants under LD.** The color code indicates the *p*-value of a two-sided Student's *t* test with Benjamini–Hochberg correction of each mutant versus the corresponding wild type. Expression levels were calculated with the ΔΔCt method using *ACT8* (AT1G49240) as a standard [81] and calibrated by biological replicate 1 (rep1) of Col-0. The data underlying this figure can be found in S1 Data and https://doi.org/10.5281/zenodo.15403194. (EPS)

**S4 Fig. Pearson's correlation analysis of transcript levels of floral regulators and different flowering traits.** Untransformed mean values of DTF, RLN and CLN in mutants **(A)** and wild types **(B)** are shown, with *p*-values shown in the upper triangles. Summarized data shown in Fig 2G. The data underlying this figure can be found in S1 Data and https://doi.org/10.5281/zenodo.15403194. (EPS)

**S5 Fig. *FLC* expression levels in parental lines of $F_2$ populations. (A)** Same data as in ref. [14]. Mean values of three replicates are shown. Pairwise comparisons, all *p*.adj. > 0.05 (Benjamini–Hochberg correction). **(B)** Crossing scheme of the *flc* mutants, flowering time (DTF), and difference in DTF. The data underlying this figure can be found in S1 Data and https://doi.org/10.5281/zenodo.15403194. (EPS)

**S6 Fig. Distribution of flowering times of F$_2$ individuals per population.** Mean values of the male and female parental lines are shown as blue and green dots, respectively. For DTF, see Fig 2B. The data underlying this figure can be found in S1 Data and https://doi.org/10.5281/zenodo.15403194.
(EPS)

**S7 Fig. Correlations of flowering trait values in all 13 F$_2$ populations.** Also shown in Fig 2D. The data underlying this figure can be found in S1 Data and https://doi.org/10.5281/zenodo.15403194.
(EPS)

**S8 Fig. QTL details. (A)**. Number of QTL per F$_2$ population and phenotype. **(B)** Total additive phenotypic variation [%] explained per population and phenotype. **(C)** Additive QTL effects from the earlier parent per F$_2$ population. Only 16 of 115 QTL showed small negative effects, with mean effects across populations: DTF −0.46, CLN −0.17, RLN −0.50, TLN −0.73. Mean effects of the other 99 QTL: DTF 1.14, CLN 0.58, RLN 1.52, TLN 2.00. **(D)** Summed QTL effects of the earlier parent per population and phenotype. Mean across populations: DTF 2.00, CLN 1.04, RLN 2.85, TLN 3.87. The respective fractions of the total effect size contributed by QTL co-localizing with *FT* are shown as green (cluster 1 *FT* QTL contributed) or red (cluster 1 *FT* QTL did not contribute) dots. **(E)** The summed QTL effects per population; all QTL effects were positive and predictive for the differences in flowering time between parental accessions (simple linear model, multiple $r^2_{adj}$ = 0.68, $p$ = 0.0005). The data underlying this figure can be found in S1 Data and https://doi.org/10.5281/zenodo.15403194.
(EPS)

**S9 Fig. Further characterization of extremely early flowering plants. (A)** Comparison of flowering time of our earliest *flc* mutants and the earliest accessions from the 1001 Genomes resource [39]. Similar letters indicate no significant difference in total leaf number (ANOVA with post hoc Tukey HSD, $p$.adj. < 0.05). **(B–K)** Field pictures of early-flowering plants in the Angit population: **(A, B)** PA4580; **(C, D)** PA4597; **(E, F)** PA3441; **(G, H)** 46U3; **(I, K)** PA3540. **(L)** Flowering time in controlled greenhouse conditions versus flowering time of wild plants: Simple linear model, multiple $r^2_{adj}$ [$p$], DTF_controlled − DTF_wild: $R^2_{adj.}$ = 0.154, $p$ > 0.05, *d.f.* = 15. The data underlying this figure can be found in S1 Data and https://doi.org/10.5281/zenodo.15403194.
(EPS)

**S10 Fig. Relative growth rates of *flc* mutants and wild types. (A)** Relative growth rate (RGR) between day 11 and 18. There were no significant differences between the mutants and wild types (Mann–Whitney *U* rank test, Benjamini–Hochberg correction, $p$.adj. > 0.05). **(B)** Ratio of RGR (RGR shown in **A**) between wild type and respective mutants (log$_2$ transformed). The colors indicate the cluster assignment as shown in Fig 4B. **(C)** PRA data of all mutant versus wild-type contrasts with a significant difference on at least one day (Mann–Whitney *U*-test rank test, Benjamini–Hochberg correction, $p$.adj. > 0.05). The data underlying this figure can be found in S1 Data and https://doi.org/10.5281/zenodo.15403194.
(EPS)

**S11 Fig. Nitrogen isotope ratios.** Nitrogen isotope ratios ($\delta^{15}$N [‰]) **(A)** and carbon to nitrogen (C/N) ratios **(B)** in *flc* mutants and wild types. Three biological replicates per line. The data underlying this figure can be found in S1 Data and https://doi.org/10.5281/zenodo.15403194.
(EPS)

**S12 Fig. Expression analyses.** Transcript analysis of *AZI1* **(A)** and *NRT1.11* **(B)** via RT-qPCR from nine-day-old plants grown under LDs. The color code indicates the $p$-value of a two-sided Student's *t* test of each mutant versus the respective wild type before and after Benjamini–Hochberg correction. Expression levels were calculated with the ΔΔCt method

using *ACT8* (AT1G49240) as a standard [81] and calibrated by biological replicate 1 (rep1) of Col-0. The data underlying this figure can be found in S1 Data and https://doi.org/10.5281/zenodo.15403194.
(EPS)

**S1 Table. Summary with all lines, flowering time, vernalization and additional data.**
(XLSX)

**S2 Table. Expression data obtained with RT-qPCR.**
(XLSX)

**S3 Table. Description of the *flc* lines used to generate F$_2$ mapping populations.**
(XLSX)

**S4 Table. Details of the genetic markers per population.**
(XLSX)

**S5 Table. Details of all detected QTL.**
(XLSX)

**S6 Table. List of flowering time genes.**
(XLSX)

**S7 Table. Details of accessions from Southern Italy.**
(XLSX)

**S8 Table. Oligonucleotide primer sequences.**
(XLSX)

**S1 Data. Figure source files and code.**
(ZIP)

## Acknowledgments

We thank members of the Weigel Lab for comments and discussion. We thank Sabine Schäfer, Natalie Betz, Alejandra Duque, Amelie-Jette Spazierer, Fabian Strauss, Nicole Vasilenko, Jonas Kreienbrink, and Amr Ziewanah for technical assistance.

## Author contributions

**Conceptualization:** Ulrich Lutz, Detlef Weigel.

**Formal analysis:** Ulrich Lutz.

**Funding acquisition:** Detlef Weigel.

**Investigation:** Ulrich Lutz, Rebecca Schwab, Wei Yuan, Marius Kollmar.

**Methodology:** Ulrich Lutz, Ilja Bezrukov, Rebecca Schwab, Wei Yuan.

**Supervision:** Ulrich Lutz, Detlef Weigel.

**Writing – original draft:** Ulrich Lutz.

**Writing – review & editing:** Ulrich Lutz, Ilja Bezrukov, Rebecca Schwab, Wei Yuan, Marius Kollmar, Detlef Weigel.

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
