## [Editor Report · Decision Letter 0]

Dear Dr Weigel, 

Thank you for submitting your manuscript entitled "Panspecific Mutation of A Flowering Regulator Reveals Cryptic Variation in Adaptive Phenotypes" for consideration as a Research Article by PLOS Biology.

Your manuscript has now been evaluated by the PLOS Biology editorial staff as well as by an academic editor with relevant expertise and I am writing to let you know that we would like to send your submission out for external peer review.

Once your full submission is complete, your paper will undergo a series of checks in preparation for peer review. After your manuscript has passed the checks it will be sent out for review. To provide the metadata for your submission, please Login to Editorial Manager (https://www.editorialmanager.com/pbiology) within two working days, i.e. by Jan 13 2025 11:59PM.

Kind regards,

Ines

--

Ines Alvarez-Garcia, PhD

Senior Editor

PLOS Biology

---

## [Decision Letter · Decision Letter 1]

Dear Dr Weigel,

Thank you for your patience while your manuscript entitled "Panspecific Mutation of A Flowering Regulator Reveals Cryptic Variation in Adaptive Phenotypes" went through peer-review at PLOS Biology. Your manuscript has now been evaluated by the PLOS Biology editors, an Academic Editor with relevant expertise, and by three independent reviewers.

As you will see, the reviewers are in general very positive and find the work novel and interesting, but they also raise several points that would need to be addressed. Reviewer 2 asks for better description of the correlations and for several clarifications, whereas Reviewer 2 also notes several issues that need to be clarified, including the methods used, and thinks that the manuscript currently over relies too much on a previous publication and that the paper should include all the details needed to allow readers to understand the results. Reviewer 1 is very positive and doesn’t have any suggestions.

In light of the reviews, we are pleased to offer you the opportunity to address the comments from the reviewers in a revision that we anticipate should not take you very long. We will then assess your revised manuscript and your response to the reviewers' comments with our Academic Editor aiming to avoid further rounds of peer-review, although we might need to consult with the reviewers, depending on the nature of the revisions.

**IMPORTANT - SUBMITTING YOUR REVISION**

3. Resubmission Checklist

a) *PLOS Data Policy*

b) *Published Peer Review*

d) *Blurb*

Please also provide a blurb which (if accepted) will be included in our weekly and monthly Electronic Table of Contents, sent out to readers of PLOS Biology, and may be used to promote your article in social media. The blurb should be about 30-40 words long and is subject to editorial changes. It should, without exaggeration, entice people to read your manuscript. It should not be redundant with the title and should not contain acronyms or abbreviations. For examples, view our author guidelines: https://journals.plos.org/plosbiology/s/revising-your-manuscript#loc-blurb

Sincerely,

Ines

--

Ines Alvarez-Garcia, PhD

Senior Editor

PLOS Biology

Reviewers' comments

Rev. 1:

This study is an extensive, and extremely well-undertaken, analysis of FLC-independent variation in flowering time in A. thaliana. This work provides a large well characterized resource for the community, important for analysis of, for example, residual vernalization sensitivities in the absence of FLC, and flowering time loci responding to metabolic signals.

An important aspect of the paper is the elaboration of the broader (and generally underappreciated) pleiotropic roles in developmental and physiological processes of FLC within the A. thaliana species. The F2 populations derived from intercrosses of 20 flc knockout mutants identified a total of 115 additive QTL. These may well help the comparative analysis of flowering time variation in other species, where the effects of FLC are also not so strong. The effects of loci linked to FLC were also clearly revealed, which had been obscured in other analyses.

This work also reinforced the conclusion that another major factor in flowering time variation in A. thaliana is FT. Their analysis predicted that extremely early accessions should be found in nature - and they validated this by prospecting for early flowering plants in Southern Italian populations. The importance of major loci in flowering time variation was not what was expected by the evolutionary biologists so the conclusions from this analysis should be of broad interest.

Previous thinking had also suggested that inactivation of FLC led many accessions towards an early flowering/fast growth strategy. However, they found flc mutants that grew more slowly, opposite to what would be expected from the prevailing slow-fast-continuum. Again, follow ups using these new resources should provide important conclusions on evolutionary trade-offs.

Rev. 2: John Lovell - note that this reviewer has signed his review

Lutz and colleagues have surveyed the multivariate phenotypic effects of FLC mutants, then explored the roles of secondary regulators within FLC mutant classes. This work predicted, then validated, that there are early flowering natural genotypes that are missed historically in collections. In addition to the direct empirical discoveries, the methods and experiments herein demonstrate the value of exploring residual variation when controlling for variation at "hub" genes; this approach could be applied to other systems and to aid the discovery of secondary loci that participate in biological processes but would otherwise be difficult to detect. Overall, I felt the text was clear and concise, the methods appropriate and well described, and the results exciting. This was a fun read! I have some comments and suggestions that I hope will improve future drafts.

John T. Lovell (18-Feb 2025)

General comments (there were no line numbers in the submission, so I pasted the text where necessary):

Throughout, correlations are interpreted as causal and indicative of pleiotropy (etc.). This is only true if they are genetic correlations (taken from the breeding values of the genotypes) and not phenotypic correlations (taken across observations including replicates within genotype). Please include descriptions including sample size and indicate which type of correlation coefficient is being reported.

Please always include the sample size, test statistic and P-value when any difference (or correlation) is reported. For example, first paragraph of p4, and many other places throughout.

There were a couple places that I had trouble following the logic in the text. In general the text is very concise, so perhaps you could add a sentence or clause in these places to flesh out how causality can be inferred in these situations:

[1] "The relative acceleration of DTF in a mutant was positively correlated with DTF of the wild type, but much weaker for RLN and CLN, which suggests that FLC affects not only time to bolting, but also the leaf initiation rate (plastochron)" — its not clear why weaker correlations demonstrate that FLC differentially affects leaf initiation rate (couldn't it just be a less heritable phenotype?)

[2] "Also as expected, residual vernalization sensitivities (expressed as log2 ratio of flowering time without and with vernalization) were higher for the knockdown lines than the complete knockouts (Mann-Whitney U rank test, p < 0.001 for both DTF and RLN), highlighting how vernalization sensitivity can be tuned by modifying FLC expression and activity" (see comment above, couldn't non-genetic variance be driving this?)

[3] "We conclude that FLC is a major component of trait de-canalization, or uncoupling" … "in contrast to the high correlations in the parental flc mutants (Fig. 1E), indicating that trait canalization due to FLC disruption". I think I get it, but it took me a while to figure out the logic about how FLC could both uncouple and couple traits.

Minor/line specific stuff:

The term "panspecific" is kind of jargon-y … I can understand that it means "all variation within a species", but the first hits for this term in a google search yielded results related to specificity of antibodies. I would strongly recommend using a different term or just describe what you mean.

I had a bit of trouble following the flow of information in the multi-panel figures. Some things that could be cleared up:

[1] The RNA-seq analysis, shown in Figure 5, is limited to seven early flc mutants, but the remainder of the analyses shown in Figure 5 (panels C-E expand) to 29 flc mutants. Why were these additional genotypes included and are they also representative of early flowering flc mutants?

[2] Figure 3A, Do the labeled arrows refer to the parents of the F2 population? If so, please indicate why not all parents used in crosses are labeled and what the significance is of the labeled genotypes.

[3] The HSD labeling in Fig 3D is a bit distracting and doesn't add much. Are you really interested in all differences, and if so, are the groupings correcting for the many multiple tests (contrasts)?

[4] Are the seven early flowering flc mutants (shown in Figure 5) represented by the Figure 3 cluster 1 earliest flowering mutants? I am missing the discussion that connects the differences in growth strategies to the FLC-independent loci that may be underlying differences in growth rates.

[5] In the text, "the difference in AZI1 was not robust to correction for multiple comparison", however the relative expression levels of AZI1 of mutants compared to wild types is denoted as highly significant as shown in Figure 5C (p ≤ 0.01; ***). See comment above about reporting of statistics.

[6] Figure 1C has dashed vertical lines that indicate 0 values … I think. Please describe in the caption.

[7] Figure 2D, its probably better to label each chromosome with the respective chromosome number (perhaps instead of the cM positions, since this is just scaled to the size of the facets).

[8] Figure 3C, Is the location of the top panel image the "Angit site"?

[9] Figure 4C, Assuming the solid, cluster specific horizontal lines are means (or medians?). Please describe in the caption.

[10] Figure 5B, The color assigned to "response to oxygen-containing compound" exceeds the "Fold enrichment" color scheme range.

There were some discrepancies between the SI figures and text:

[1] Figure S9 indicates that there may be haplotype-specific structural variation at the FT locus, but this figure or insight is not discussed in the main text. Also, for pane B of this supplemental figure, not all F2 populations are represented, which would be useful to establish the existing variation at the FT locus for all parents of F2 populations used in QTL mapping. For the other six F2 families not represented in pane B, did the parents not have a QTL that colocalized in the FT genomic region?

[2] The text for Figures S10 and S11 is included in the file, but the figures are missing.

[3] Figure S8, C) please indicate the green vertical line as positions of "a priori candidate genes" as denoted in the main text D) the blue and pink dots are referred to as "green" and "red".

Rev. 3:

This paper focuses on the genetic basis of flowering time variation in Arabidopsis following mutation of one of the main flowering regulators, FLC. Genetic mapping when FLC is perturbed reveals new flowering loci and expands the range of flowering time diversity (and other phenotypes) in this species. This work is pretty neat and will likely be of interest to researchers in the Arabidopsis community and potentially beyond.

While I enjoyed reading this paper, there were several major concerns:

-Some terminology is used that is not common in genetics. What are panspecific mutants or prognostic sequence analysis? I would recommend avoiding such jargon and writing in simple language.

-Understanding this manuscript, especially the panspecific mutants component, requires reading at least one prior paper. This current paper needs to be able to entirely stand on its own. A reader should not be expected to read another paper to be able to understand the current one. In the main text, the nature of the mutations and the accessions carrying them should be explicitly described. So much depends on that information and it is critical to make it simple to understand.

-More description is needed in the main text about the crosses. Which accessions/mutants are being crossed? Ideally there would be a figure or panel summarizing this in an easy-to-understand way. I found some of this information in the Supplement, but if asked, I could not fully explain the exact experimental design underlying this work.

-Connected to the above, the FLC region comes up as a locus in numerous crosses. If some of the mutants are just knockdowns, couldn't detection of the FLC locus simply reflect variation at FLC itself? This is a finding that connects back to the importance of clearly explaining the experimental design. If it is possible that FLC may itself be the QTG at the QTL, that would be good to present as an alternative explanation.

-The final section of the Results seemed tacked on. The prior section about the Italian strains from Angit and the detection of part of the phenotypic spectrum that was missed was compelling and a thoughtful extension of the preceding sections. However, the final Results section did not integrate as well with the other work. Maybe it can be better explained how the final presented results integrate with everything else. I did not think the final Results section was critical to the main thrust of the paper, but maybe I misunderstood it.

- Generally, the IDs used in figures are not easy to understand. There are many main text and supplemental panels containing theses IDs. What accessions/mutants are being examined in each figure? Connected to this, often the labels are too small or cramped to easily read (e.g., Figs S3, S7, S9, S13).

-Figs S10 and S11 were absent from the document I could access. I downloaded multiple times, but they were absent every time, suggesting they might be missing from the submission.

There was also a minor typo:

-In second sentence of introduction, delete 'it' in 'Flowering it is orchestrated….'

---

## [Editor Report · Decision Letter 2]

Dear Dr Weigel,

Thank you for your patience while we considered your revised manuscript entitled "Species-Wide Gene Editing of A Flowering Regulator Reveals Hidden Phenotypic Variation" for publication as a Research Article at PLOS Biology. This revised version of your manuscript has been evaluated by the PLOS Biology editors and by the original Academic Editor.

Based on our Academic Editor's assessment of your revision, we are likely to accept this manuscript for publication, provided you satisfactorily address the data and other policy-related requests stated below.

We expect to receive your revised manuscript within two weeks. 

*Published Peer Review History*

*Press*

Sincerely,

Ines

--

Ines Alvarez-Garcia, PhD

Senior Editor

PLOS Biology

DATA POLICY:

2) Deposition in a publicly available repository - you could add them to Zenodo via GitHub (see my note about our Code policy below). Please also provide the accession code or a reviewer link so that we may view your data before publication. 

Fig. 1A-D; Fig. 2A-F; Fig. 3D; Fig. 4A-D; Fig. 5A-E; Fig. S1A-C; Fig. S2A-F; Fig. S3; Fig. S4A, B; Fig. S5A, B; Fig. S6A, B; Fig. S7A-C; Fig. S8A-E; Fig. S9A, L; Fig. S10A-C; Fig. S11A, B and Fig. S12A, B

Please also ensure that figure legends in your manuscript include information on WHERE THE UNDERLYING DATA CAN BE FOUND - if you add the data to Zenodo, you should add the link in the corresponding figure legends, and please label the data according to the figures to make it easier to readers.

Please also ensure that your Data Statement in the submission system accurately describes where your data can be found.

FINANCIAL DISCLOSURE

Please complete the Financial Disclosure statement with the following details:

• Initials of the authors who received each award

• Grant numbers awarded to each author

• The full name of each funder

• URL of each funder website

• Did the sponsors or funders play any role in the study design, data collection and analysis, decision to publish, or preparation of the manuscript?”

CODE POLICY

Thank you for depositing the code in GitHub

---

## [Editor Report · Decision Letter 3]

Dear Dr Weigel,

Thank you for the submission of your revised Research Article entitled "Species-Wide Gene Editing of A Flowering Regulator Reveals Hidden Phenotypic Variation" for publication in PLOS Biology. On behalf of my colleagues and the Academic Editor, Leonie Moyle, I am delighted to let you know that we can in principle accept your manuscript for publication, provided you address any remaining formatting and reporting issues. These will be detailed in an email you should receive within 2-3 business days from our colleagues in the journal operations team; no action is required from you until then. Please note that we will not be able to formally accept your manuscript and schedule it for publication until you have completed any requested changes.

PRESS

Sincerely, 

Ines

--

Ines Alvarez-Garcia, PhD,

Senior Editor

PLOS Biology
